# CoNiCrMo Particles, but Not TiAlV Particles, Activate the NLRP3 Inflammasome in Periprosthetic Cells

**DOI:** 10.3390/ijms24065108

**Published:** 2023-03-07

**Authors:** Fenna Brunken, Tristan Senft, Maria Herbster, Borna Relja, Jessica Bertrand, Christoph H. Lohmann

**Affiliations:** 1Department of Orthopaedic Surgery, Otto-von-Guericke-University, 39120 Magdeburg, Germany; 2Institute of Materials and Joining Technology, Otto-von-Guericke-University, 39120 Magdeburg, Germany; 3Department of Experimental Radiology, Otto-von-Guericke-University, 39120 Magdeburg, Germany

**Keywords:** inflammasome, wear debris, endoprosthesis

## Abstract

Aseptic loosening is the main reason for arthroplasty failure. The wear particles generated at the tribological bearings are thought to induce an inflammatory tissue response, leading to bone loss and the subsequent loosening of the implant. Different wear particles have been shown to activate the inflammasome, thereby contributing to an inflammatory milieu in the direct vicinity of the implant. The aim of this study was to investigate whether the NLRP3 inflammasome is activated by different metal particles in vitro and in vivo. Three different cell lines representing periprosthetic cell subsets (MM6, MG63 and Jurkat) were incubated with different amounts of TiAlV or CoNiCrMo particles. The activation of the NLRP3 inflammasome was determined through the detection of the caspase 1 cleavage product p20 in a Western blot. The formation of the inflammasome was also investigated in vivo using immunohistological staining for ASC in primary synovial tissues as well as tissues containing TiAlV and CoCrMo particles and in vitro after the stimulation of the cells. The results show that the CoCrMo particles induced ASC more markedly, as a readout for inflammasome formation in vivo, compared to TiAlV particular wear. The CoNiCrMo particles also induced ASC-speck formation in all the tested cell lines, which was not induced by the TiAlV particles. The Western blot shows that NRLP3 inflammasome activation, measured through caspase 1 cleavage, was increased only by the CoNiCrMo particles in the MG63 cells. We conclude from our data that the activation of the inflammasome is mainly driven by CoNiCrMo particles and less by TiAlV particles, indicating that different inflammatory pathways are activated by the different alloys.

## 1. Introduction

The aseptic loosening of endoprosthetic implants occurs in about 15–20% of all patients within 20 years after joint replacement (TJR) [1]. Due to the increasing number of TJR implantations, especially in younger patients, it is thought that the problem of aseptic loosening will increase in the future [2,3,4]. The most common revision diagnoses in total knee replacements are loosening (39.9%), infection (27.4%), instability (7.5%), a periprosthetic fracture (4.7%) and arthrofibrosis (4.5%) [1]. Further studies indicated that aseptic loosening is the most frequent cause for prosthesis revisions [5,6].

One main reason for aseptic implant loosening is the wear particles that are liberated from the articulating surface which are taken up by periprosthetic cells [7]. Primarily responsible for aseptic prosthesis loosening are abrasive particles accumulating in the tissue and inducing inflammation; subsequent osteolysis; and, therefore, implant loosening [8]. The cellular responses to metal debris might depend on the size, shape, alloy type or number of particles [9,10,11]. One pathway, however, that is directly linked to particle-induced inflammation is the inflammasome pathway [12]. The inflammasome pathway is not fully investigated with respect to the aseptic loosening of TJR, but there are indications that inflammasome-dependent inflammation is involved in osteolysis [13].

The inflammasome pathway has been discovered to be the main cause of inflammation during gout [14]. The NLRP3 inflammasome in gout is activated by uric acid crystals, which lead to a disruption in the lysosomal membrane and a release of cathepsin B, increasing the formation of the NLRP3 inflammasome [15]. The inflammasome is a cytosolic protein complex that induces the release of proinflammatory and cytotoxic cytokines upon activation. The NLRP3 inflammasome consists of NLRP3 (NACHT, LRR and PYD domains containing protein 3), caspase 1 and ASC (apoptosis-associated speck-like protein containing a caspase recruitment domain). The activation of the inflammasome converts procaspase 1 into active caspase 1, which subsequently cleaves pro-IL-1β (interleukin-1 beta) into its active form and therefore leads to an inflammatory tissue reaction [16,17]. Previous studies have shown that different metal ions and particles can activate the NLRP3 inflammasome in macrophages [18,19]. It has also been demonstrated that the inhibition of the NLRP3 inflammasome can reduce IL-1β secretion by macrophages after incubation with Ti (titanium) particles [19].

The aim of this study was to investigate whether the inflammasome is activated in the periprosthetic tissue in vivo and whether there is a difference between CoNiCrMo and TiAlV particles.

## 2. Results

### 2.1. Inflammasome Activation in Periprosthetic Tissue of Patients with CoCrMo Alloy Implants Increased Compared to TiAlV Implants

First, we investigated whether the inflammasome is activated in the synovial and periprosthetic tissues of patients undergoing TJR revision surgery. The representative images in Figure 1A show ASC staining (red) in the periprosthetic tissue of the TJR revision patients and the synovial tissue of the patients undergoing primary endoprosthesis implantation. The periprosthetic tissue from the revision surgeries was separated into the two groups of TiAlV (Titanium-aluminum-vanadium) or CoCrMo (cobalt-chromium-molybdenum) implants. Interestingly, we found increased ASC staining in the periprosthetic tissue from the patients with the visible wear debris of either CoCrMo/polyethylene (PE) or from the patients with TiAlV/PE pairings. However, only the tissues containing the CoCrMo wear particles showed a statistically significant increase in ASC (*p* = 0.004) compared to the control tissue. The difference between the Ti particle containing tissue and control tissue was not statistically significant (*p* = 0.12) (Figure 1B). When we plotted the implantation duration of the Ti and CoCrMo implants prior to the revision surgery against the amount of ASC staining in the tissue, we observed an increased ASC presence in the CoCrMo wear tissue with an increasing implantation time (r^2^ of 0.68, *p* = 0.02), which was not detected in the periprosthetic tissue containing Ti wear (r^2^ of 0.45, *p* = 0.1) (Figure 1C). This finding prompted us to investigate the differential effect of CoCrMo and TiAlV particles on the activation of the inflammasome in different periprosthetic cell lines.

### 2.2. Characterization of Particles Used for the In Vitro Experiments

To further investigate the effect of the CoCrMo and TiAlV particles on the activation of the NLRP3 inflammasome, we generated particles from the CoNiCrMo (cobalt-nickel-chromium-molybdenum) and TiAlV alloys. The particles were examined for their chemical composition and morphology using SEM-EDX (scanning electron microscopy – energy-dispersive X-ray spectroscopy). Figure 2A shows a representative picture of the CoNiCrMo particles, and Figure 2B shows a representative electron micrograph of the TiAlV particles. The EDX analysis of the particles shows that only the elements of the respective alloys were present, and no contaminating elements were found. The particle composition shows the quantitative composition of the nominal alloys (Figure 2C,D). Both the particle types showed a great diversity in shape and size, with an irregular shape and a predominantly rough surface. The exact particle size distribution was determined through laser diffraction using the Mastersizer 3000E. In the CoNiCrMo solution, the volume fraction of the particles with a size > 100 µm was significantly larger than that in the TiAlV solution. In both the solutions, the particles with a size of approx. 10 µm made up the largest part of the volume. In the size range of < 10 µm, both types of particles showed an almost identical size distribution (Figure 2E,F). 

### 2.3. Intracellular Uptake of Particles Did Not Influence the Cell Viability

Monocytes (MM6), osteoblasts (MG63) and T cells (Jurkat) were incubated with different amounts (10^5^–10^7^) of particles for 24 h and were stained with phalloidin and DAPI to visualize the cytoskeleton and nucleus and, therefore, the uptake of the particles into the cells (Figure 3A). The representative pictures show that the TiAlV (TAV) particles as well as the CoNiCrMo particles were taken up by all the tested cells. 

To determine whether the particles used in the experiments had a direct cytotoxic effect on the cells, a WST-assay was performed. The test showed that the particles did not significantly influence the viability of the cells in the applied concentrations (Figure 3A–C). 

### 2.4. Increased ASC-Speck Formation upon Stimulation with CoNiCrMo Particles 

To investigate whether there are differential effects of the two tested metal particle types on the formation of the inflammasome, we treated the monocytes (MM6), osteoblasts (MG63) and T cells (Jurkat) with different concentrations of the particles (10^5^, 10^6^ and 10^7^). We used ASC immunostaining to visualize the speck formation, which is a critical step in the formation of the inflammasome. We quantified the ASC-speck-positive cells at each particle concentration. The representative images for each condition are shown in the left panel of Figure 4. We observed an approximately 4-fold increase in ASC-speck-positive monocytes (MM6) with the CoNiCrMo particles at the concentration of 10^6^ in comparison to the untreated control (*p =* 0.04) (Figure 4A). Additionally, for the MG63 osteoblast cell line, a significant increase in ASC-speck-positive cells was observed at a concentration of 10^5^ of the CoNiCrMo particles (*p =* 0.02) (Figure 4B). In the T cells, a concentration of 10^5^ of the CoNiCrMo particles induced a significant increase in ASC-speck formation (Figure 4C). 

Interestingly, the TiAlV particles did not induce ASC-speck formation in the MM6, MG63 or Jurkat cells at any concentration.

### 2.5. CoNiCrMo Particles Induced Inflammasome-Dependent Signaling in MG63 Cells

To evaluate the effect of both particle types on the activation of the inflammasome, we investigated the activation of caspase 1 by detecting the p20 cleavage fragment of caspase 1 using Western blotting. The Western blot showed a significant increase in the p20 of caspase 1 in the MG63 cells after stimulation with a concentration of 10^7^ of the CoNiCrMo particles (*p* = 0.04) (Figure 5B). The TiAlV particles did not induce caspase 1 cleavage in the MG63 osteoblastic cells. The cleavage of caspase 1 in the MM6 and Jurkat cells was not changed through incubation with either the TiAlV or CoNiCrMo particles.

## 3. Discussion

Previous studies have shown that the NLRP3 inflammasome can be activated by metal particles from orthopedic implants [18,20,21]. Most studies, however, have focused on inflammasome activation in macrophages, whereas the contribution of other periprosthetic cells on inflammasome-dependent inflammation is not well understood. Interestingly, there is an association between the NLRP3 and CARD8 inflammasome polymorphisms and aseptic implant loosening [22].

We observed ASC-speck formation as an indicator for inflammasome activation in MM6 monocytes, MG63 osteoblasts and Jurkat T cells after stimulation with CoNiCrMo particles, indicating that a broad range of periprosthetic cells are capable of inflammasome formation in vitro. This is in line with another study showing that NLRP3 is expressed in various cells, e.g., neutrophils, monocytes, dendritic cells, T cells, B cells and osteoblasts, indicating their ability for inflammasome formation [23]. However, we observed the activation of the NLRP3 inflammasome and the subsequent cleavage of caspase 1 only in the osteoblastic cell line (MG63) upon stimulation with the CoNiCrMo particles. The marked effect of CoCrMo particles on macrophage-dependent NLRP3 inflammasome activation has been described [18]. Interestingly, this study shows that the inflammasome can be exclusively activated by CoNiCrMo particles but not TiAlV particles in osteoblasts. This is in contrast to another study showing that titanium (Ti) particles were able to activate the NLRP3 inflammasome components in macrophages [24].

Although it has been shown that the inflammasome can be activated by wear debris, it is still controversial whether metal particles alone are able to prime the inflammasome (e.g., via DAMP pathways) or if an additional stimulus is needed for this. The role of exogenous PAMPs and TLRs in this process remains especially unclear [24,25]. There are different signals that can lead to the activation of the NLRP3 inflammasome, e.g., extracellular ATP leading to an efflux of potassium, lysosomal disruption or reactive oxygen species [17,26]. It has been shown that the release of IL-1β by macrophages caused by Ti and CoCrMo particles depends on the release of cathepsin B into the cytosol [20,27]. This indicates that the inflammasome is activated through the destabilization of the lysosome after the phagocytosis of the particles. Metal debris < 10 µm probably causes a stronger proinflammatory response because of the high ability of cells to phagocytize these particles [28]. It was shown that irregular particles smaller than 6 µm probably induce the activation of the inflammasome in macrophages independent of particle phagocytosis, possibly due to an increased release of metal ions from irregular particles compared to smooth particles [27]. Furthermore, the size and shape of the particles might have influenced the cellular reaction towards the CoNiCrMo and TiAlV particles in this study. The particle size used in vitro in this study increased in comparison to the size distribution of the wear debris found in the in vivo total joint replacements [8]. Furthermore, there was a difference in size between the CoNiCrMo particles, which were partially bigger than the TiAlV particles used in this study.

Co particles have been shown to cause more inflammatory responses, such as HIF-1α expression and ROS production, in the cells compared to the Ti particles, which might explain the difference in the activation of the inflammasome in osteoblasts [25,29]. This observation is in line with the findings of this study, indicating a much stronger in vivo and in vitro effect of the Co containing particles.

## 4. Materials and Methods

### 4.1. Patient Samples

Patients undergoing TJR or TJR revision surgery were included in this study. Patients with macroscopically visible metallic wear particles from either CoCrMo-PE pairings or TiAlV-PE pairings were included. Informed consent was obtained from patients prior to inclusion in this study. Institutional Review Boards at the University of Magdeburg and the University of Tartu approved the protocols prior to commencement of the study (IRB No 150/12 and Tartu No 227/T-14). 

### 4.2. Immunohistochemistry of ASC

Paraffin sections from synovial tissue from primary and revision surgeries were deparaffinized and rehydrated. For antigen retrieval, sections were pretreated with citrate buffer at pH of 9 for 1 h. Free epitopes were blocked with 4% BSA in PBS for 1 h at room temperature. Tissue sections (3–4 µm) were stained with primary antibody against ASC (sc-514414, Santa Cruz Biotechnology, Dallas, TX, USA, 1:100). Alexa Fluor^®^ 555 (Thermo Scientific, Waltham, MA, USA) was applied as secondary antibody. Sections were fixed with Roti^®^-Mount FluorCare DAPI (Carl Roth, Karlsruhe, Germany). Images were acquired at 22 °C with Zeiss Axiovert microscope (Zeiss, Oberkochen, Germany) using 40× magnification/0.75 numerical aperture using Zen 3.0 software.

### 4.3. Cell Culture

Monocyte (MM6), T cell (Jurkat) and osteoblast (MG63) cell lines were stimulated for 24 h with different amounts of CoNiCrMo and TiAlV particles. A total amount of 10^5^, 10^6^ or 10^7^ particles was used per well in a 24-well plate. The particles were counted using a Thoma cell counting chamber. The cells were incubated for 24 h at 37 °C and 5% CO2. Afterwards, the metal particles were added, and the cells were incubated for another 24 h. Jurkat and MM6 cells were cultured in RPMI 1640 medium (Biochrom, Schaffenhausen, Switzerland). MG63 cells were cultured in DMEM medium (Biochrom, Schaffenhausen, Switzerland). All cell culture media contained 1% penicillin/streptomycin (Biochrom Schaffenhausen, Switzerland,) and 10% heat-inactivated FCS (PAN, Aidenbach, Germany). 

For the WST-assay, 10^4^ cells were seeded into 96-well plates with 100 µL of medium. For the Western blot, 3 × 10^5^ MG63 cells were seeded into 12-well plates, and 1 × 10^6^ suspension cells (MM6 and Jurkat) were placed in 24-well plates. A total of 1000 µL of medium was added to each well. For the immunocytochemistry and phalloidin-red staining, the suspension cells were also placed in 24-well plates (1 × 10^6^ cells/well), and MG63 cells were placed in 48-well plates (0.2 × 10^5^ cells/well); 500 μL of medium was added to each well. 

### 4.4. Particle Analysis

The particles were generated from the following biomedical implant materials: Co-35Ni-20Cr-10Mo (CoNiCrMo, ISO 5832-6, ASTM F562) and Ti-6Al-4V (TiAlV, ISO 5832-3, ASTM F1472) alloys. Briefly, metal bars of one alloy were filled into containers made from the same material, and absolute ethanol was added. Rotation caused abrasion due to the contact of the metal bars and the containers. The particles were filtered using 0.1 µm polyester PETE Membrane Filters (Sterlitech, Auburn, WA, USA), and particles with a diameter lager than 0.1 µm were used for the experiments. The particles were transferred to a carbon pad and were sputter-coated with gold before the SEM analysis. The composition and size distribution of the particles were analyzed through SEM (FEI XL30 FEG ESEM, FEI/Philips, Amsterdam, Nethderlands) and with the Mastersizer 3000E (Malvern Panalytical, Kassel, Germany).

### 4.5. WST-Assay

To assess the cell viability, after particle stimulation, a WST-1 assay was performed according to the manufacturer’s protocol. The amount of formazan was measured at a wavelength of 430 nm using the Tecan infinite F200 pro (Tecan, Maennedorf, Switzerland). 

### 4.6. Cytoskeleton Staining

The cells were treated with metal particles as described above; after 24 h of stimulation, cells were trypsinized, and the suspension cells were transferred to 48-well plates containing coverslips pretreated with poly-L-Lysine and were incubated for 30 min at 37 °C. The MG63 cells were cultured in 48-well plates that already contained coverslips. The cells were fixed with 500 µL of 4% PFA/well for 30 min at room temperature and were rinsed 3 times with PBS. Cells were stained with Alexa Fluor 555 Phalloidin (#8953Cell Signaling, Leiden, Netherlands) for visualization of the actin cytoskeleton. The coverslips were incubated for 15 min at room temperature in the dark and were rinsed again with PBS before they were covered with DAPI. The staining was evaluated using an Axio Observer Z1 microscope with an Axiocam 702 mono camera, and the stained area per cell was measured using ImageJ 1.

### 4.7. Immunocytochemistry

The cells were fixed with a methanol–acetone mixture (500 µL of 1:1 methanol–acetone mixture was added to each well for 5 min at 4 °C) and were permeabilized with Triton (500 µL of 0.1% Triton in PBS were added to each well for 10 min at room temperature). After rinsing the coverslips with PBS, the cells were incubated with 4% BSA in PBS for 1 h at room temperature. The coverslips were incubated with the antibody against ASC (sc-514414, Santa Cruz Biotechnology, 150 µL, 1:500) overnight at 4 °C. After 24 h, the coverslips were rinsed again with PBS and were incubated with the secondary antibody for 1 h in the dark (AF555 anti-mouse, 1:200, 150 µL/well). The coverslips were rinsed with PBS, stained with DAPI and mounted on microscope slides. The staining was evaluated using an Axio Observer Z1 microscope with an Axiocam 702 mono camera (Zeiss, Oberkochen, Germany), and the stained area per cell was measured using ImageJ. 

### 4.8. Western Blot

The cells were washed with PBS and were then lysed with NP40-buffer for 1 h at 4 °C on a roller shaker and were centrifuged at 12.000× *g* for 10 min. The protein concentration was measured using the Pierce BCA Protein Assay Kit (Thermo Scientific, Waltham, MA, USA). The samples were diluted to a concentration of 20 µg of protein in a volume of 20 µL. The samples were separated in a 10% acrylamide gel and were transferred to a nitrocellulose membrane. After blocking the membranes with 5% BSA for 1 h at room temperature, they were incubated overnight at 4 °C with the primary antibody against caspase 1 (dilution 1:400, Abcam, ab-54932). The membranes were washed 3 × 10 min with TBS-T, were incubated with the secondary antibody (goat anti-mouse IgG HRP, 1:10.000, Santa Cruz Biotechnology) for 1 h at room temperature and were washed again with TBS-0.5% Tween. The proteins were detected with a Clarity Western ECL solution. After stripping the membranes, they were stained with the anti-GAPDH monoclonal mouse antibody (1:500, Cell Signaling Technology, Danvers, MA, USA) and with the secondary antibody (goat anti-mouse IgG HRP, 1:10.000, Santa Cruz Biotechnology Dallas, TX, USA) as described above. 

### 4.9. Statistical Analysis

The data were analyzed using GraphPad Prism. For multiple comparisons, the Friedman test (ANOVA) with Dunn’s post hoc correction was used. *p*-values < 0.05 were considered statistically significant. 

## 5. Conclusions

Our data indicate that the inflammasome-driven inflammatory reaction towards wear debris is mainly caused by CoCrMo particles, whereas TiAlV particles do not seem to trigger this pathway.

## Figures and Tables

**Figure 1 ijms-24-05108-f001:**
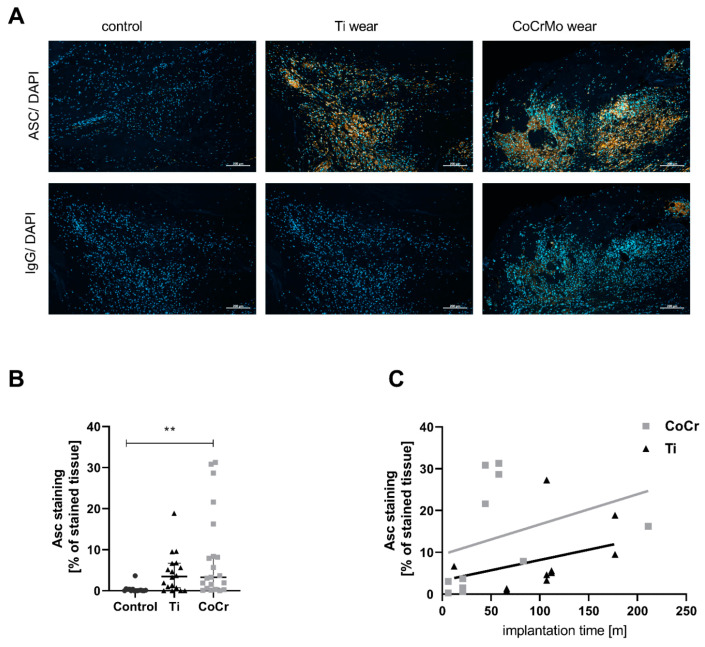
Inflammasome activation in periprosthetic tissue of patients with CoCrMo alloy implants increased compared to TiAlV implants. (**A**) Representative pictures of a periprosthetic tissue with a CoCrMo or TiAlV implant as well as primary tissue (ASC is red, and DAPI is blue). The bottom panel shows the corresponding IgG control staining. Scale bar 200 µm. (**B**) Quantification of ASC stained tissue area for control (primary, N = 18), tissue with TiAlV particles (Ti, N = 21) and tissue with CoCrMo particles (CoCr, N = 15). The data were analyzed using a one-way ANOVA with Holm–Sidak post hoc correction for multiple testing (F (2, 51) = 5.30, *p* = 0.0081). The difference for control vs CoCr was statistically significant (*p =* 0.004). (**C**) Correlation between implantation time and the percentage of ASC stained tissue area for TiAlV wear and CoCrMo wear containing tissue. Spearman correlation analysis indicated a correlation for CoCrMo with the implantation time (r^2^ of 0.70, 95% CI: 0.15 to 0.92, *p =* 0.02). The correlation for Ti with the implantation time was not significant (r^2^ of 0.45, 95% CI: −0.18 to 0.82, *p =* 0.14). ** *p* > 0.01.

**Figure 2 ijms-24-05108-f002:**
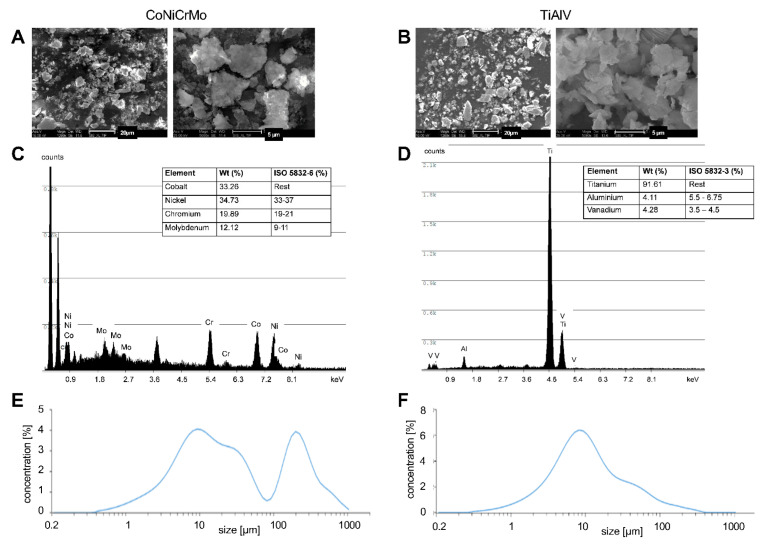
Characterization of particles used for the in vitro experiments. (**A**) SEM image of the CoNiCrMo particles at 1200× magnification and at 5000× magnification. (**B**) SEM image of the TiAlV particles at 1200× magnification and at 5000× magnification. (**C**) EDX spectrum of the analyzed CoNiCrMo particles. (**D**) EDX spectrum of the analyzed TiAlV particles. (**E**) Size distribution of CoNiCrMo and (**F**) TiAlV particles measured with the Mastersizer 3000E.

**Figure 3 ijms-24-05108-f003:**
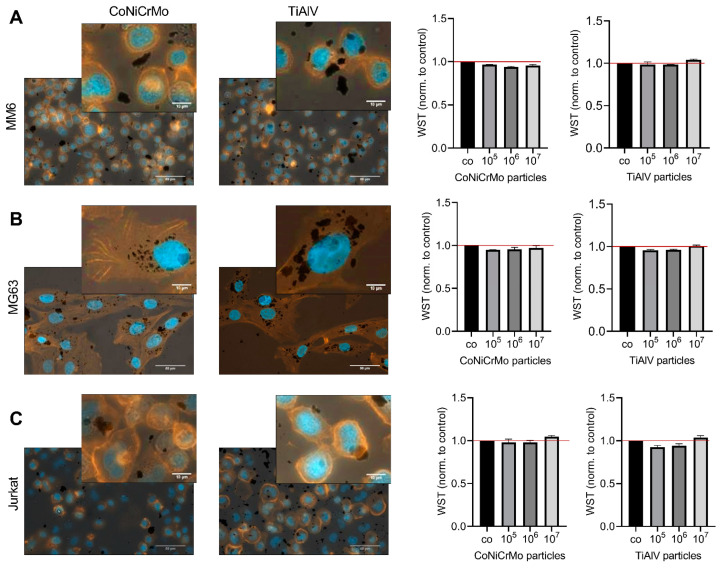
Intracellular uptake of particles did not reduce the cell viability. (**A**) Phalloidin staining (red) of the cytoskeleton of monocytes (MM6), (**B**) osteoblasts (MG63) and (**C**) T cells (Jurkat) with bright field overlay picture to visualize the particles. The influence of CoNiCrMo particles as well as TiAlV particles at different concentrations on the cell viability (WST-assay) was measured after 24 h. The data were analyzed using Friedman test and Dunn’s correction for multiple testing. No significant differences in the WST-assay were observed under the tested conditions (N = 3). Scale bar 10 µm for all iamges.

**Figure 4 ijms-24-05108-f004:**
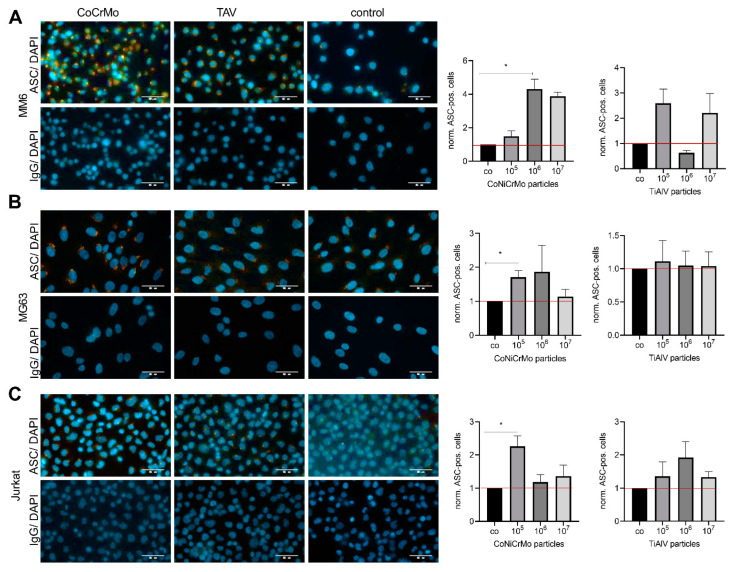
Increased ASC-speck formation upon stimulation with CoNiCrMo particles. Quantification of speck-positive MM6 (**A**), MG63 (**B**) and Jurkat (**C**) cells after treatment with different concentrations (10^5^, 10^6^ and 10^7^) of CoNiCrMo or TiAlV particles compared to nontreated control. Representative pictures for each are shown on the right-hand side (ASC is red, and DAPI is blue). The quantification of speck-positive cells after incubation with either CoNiCrMo or TiAlV particles normalized to the untreated controls (co) is given on the left side. The statistical evaluation was performed using a Friedman test with Dunn’s post hoc correction for multiple testing (N < 3). * *p* > 0.05. Scale bar 50 µm.

**Figure 5 ijms-24-05108-f005:**
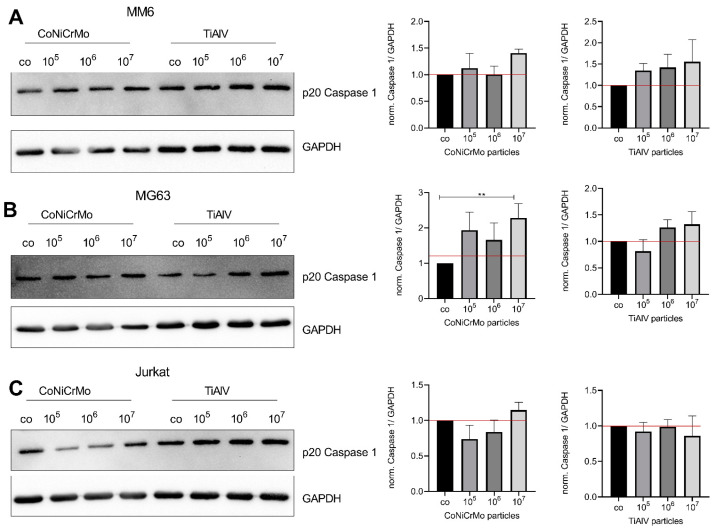
CoNiCrMo particles induced inflammasome-dependent signaling in MG63 cells. Representative Western blots of MM6 (**A**), MG63 (**B**) and Jurkat (**C**) cells stimulated with different concentrations of CoNiCrMo and TiAlV particles for the activation of caspase 1 (p20 of caspase 1) and of the corresponding loading control (GAPDH) are given in the right panel. The quantification of cleaved caspase 1 (p20) normalized to the untreated control (co) is depicted in the graphs on the left-hand side. The data were analyzed for statistical significance using a Friedman test with Dunn’s post hoc correction for multiple testing (N ≤ 3). *** p* < 0.01.

## Data Availability

The data presented in this study are available on request from the corresponding author. The data are not publicly available due to ethical reasons.

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
