# Peer review of "CoNiCrMo Particles, but Not TiAlV Particles, Activate the NLRP3 Inflammasome in Periprosthetic Cells"

_ijms, 2023, doi:10.3390/ijms24065108_

Round 1
Reviewer 1 Report
The manuscript by Jessica Bertrand and co-workers, describes the activation of NLRP3 inflammasome by CoNiCrMo and TiAlV using both in vitro and in vivo assays. Overall, the manuscript is well structured including a substantial amount of work. However, it is necessary to consider the following observations before accepting the manuscript.
1.-In the tests carried out on the cells (MM6, Jurkat and MG63), how were the CoNiCrMo and TiAlV particles administered to the cells?
2.-In the assays performed on the cells (MM6, Jurkat and MG63), were any positive controls (drug or active compound previously reported) included?
3.-It is recommended to include a conclusion.
Author Response
We would like to thank the reviewer for the comments and will answer the questions on a point to point basis.
1.-In the tests carried out on the cells (MM6, Jurkat and MG63), how were the CoNiCrMo and TiAlV particles administered to the cells?
The particles were counted using a cell counting chamber. The amount of 105, 106 and 107 particles resuspended in cell culture medium were given to the cell culture.
We have tried to clarify this procedure in the methdos section.
2.-In the assays performed on the cells (MM6, Jurkat and MG63), were any positive controls (drug or active compound previously reported) included?
We did not use any drugs as positive control.
3.-It is recommended to include a conclusion
Thank you for making us aware that the header for the conclusion section got lost during formatting. We have included the header again. The conclusion is:
Our data indicate that the inflammasome driven inflammatory reaction towards wear debris is mainly caused by CoCrMo particles, whereas TiAlV particles do not seem to trigger this pathway.
Reviewer 2 Report
The authors carried out experiments indicating that the inflammasome driven inflammatory reaction towards wear debris is mainly caused by CoCrMo particles, whereas TiAlV particles do not seem to trigger this pathway. That research topic is unfortunately far away from my experiences, but nevertheless it sounds interesting and the experiments seem to be done carefully. So I cannot really assess the novelty of that work. Moreover, I am not convinced that the authors have chosen IJMS as the proper journal. So I would would like to recommend to resubmit this manuscript to a more specialized journal.
Just a minor point: line 190: "EDS spectra" change by "EDS analysis", because EDS means already "Energy-dispersive X-ray Spectroscopy"
Author Response
We would like to thank the reviewer for the comments. We have changed the term EDS spectra to EDS analysis.
Reviewer 3 Report
The authors have tested two types of nano materials CoCrMo particles and lTiAlV particles on their ability to activate the inflammasome through ASC- Speks and Caspase markers.
1. In abstract during describing results you have mentioned CoCrMo instead of CoNiCrMo, please do recheck. Also confirm that you have mentioned CoCrMo and CoNiCrMo correctly in the appropriate places
2. There are some grammar mistakes in the manuscript, please do recheck.
3. Other studies shows that Ti particles can able to activate NLRP3 inflammasome in your result you have mentioned that Ti particles could not able to activate please give some supportive reason why it couldn’t occur here, so that it could justify your study.
4. In methodology, the cell numbers were given in wrong format
5. Line No. 9 grammar Correction, the line “Aseptic loosening main reasons for arthroplasty failure”
6. In figures 4 and 5, theTIAlV seems to induce ASC levels, but not significantly, but caspase levels are significantly upregulated in both treatments except jurkat cells. What is authors conclusion for this observation?
7. “Interestingly, this study shows that the inflammasome can be exclu- 278 sively activated by CoNiCrMo but not TiAlV particles in osteoblasts. This is in contrast to 279 another study showing that titanium (Ti) particles were able to activate NLRP3 in- 280 fammasome components in macrophages”
– What we can observe in p20 Caspase 1 form figure 5 is, Ti group had upregulated protein at 10^7, than the CoNiCrMo groups, please reconsider this statement, as the results are suggesting the other way.
Author Response
We would like to thank the reviewer for the comments and will answer the questions on a point to point basis.
- In abstract during describing results you have mentioned CoCrMo instead of CoNiCrMo, please do recheck. Also confirm that you have mentioned CoCrMo and CoNiCrMo correctly in the appropriate places
We are sorry for this mistake. We have corrected CoCrMo in the abstract to CoNiCrMo. The in vivo detected particles were indeed CoCrMo instead of CoNiCrMo, because there was a ban of Ni in endoprosthesis and no Ni is included in the endoprosthesis alloys anymore.
- There are some grammar mistakes in the manuscript, please do recheck.
We thank the reviewer for making us aware of this fact and have rechecked the manuscript.
- Other studies shows that Ti particles can able to activate NLRP3 inflammasome in your result you have mentioned that Ti particles could not able to activate please give some supportive reason why it couldn’t occur here, so that it could justify your study.
As stated in our discussion: ….”]. It was shown that irregular particles smaller than 6µm probably induce activation of the inflammasome in macrophages independent of particle phagocytosis, possibly due to an increased release of metal ions from irregular particles compared to smooth particles [27]. Furthermore, the size and shape of the particles might have influenced the cellular reaction towards CoNiCrMo and TiAlV particles in this study. The in vitro used particle size in this study is increased in comparison to the size distribution of wear debris found in vivo total joint replacements [8]. Furthermore, there is a difference in size between CoNiCrMo particles, which are partially bigger than the TiAlV particles used in this study.” We think that the particle size might influence the uptake and thereby the activation of the inflammasome. In other studies the particle size might have been different and therefore the cell response might have been different.
- In methodology, the cell numbers were given in wrong format#
We are sorry for this mistake and have corrected the cell number.
- Line No. 9 grammar Correction, the line “Aseptic loosening main reasons for arthroplasty failure”
We would like to thank the reviewer for bringing this mistake to our attention. We have corrected the sentence. It now reads: Aseptic loosening is the main reasons for arthroplasty failure.
- In figures 4 and 5, theTIAlV seems to induce ASC levels, but not significantly, but caspase levels are significantly upregulated in both treatments except jurkat cells. What is authors conclusion for this observation?
We are sorry for the misunderstanding. The caspase 1 cleavage is only significantly induced in MG63 cells incubated with CoNiCrMo particles. This is also stated in the manuscript text: TiAlV particles did not induce caspase 1 cleavage in MG63 osteoblastic cells. The cleavage of caspase 1 in MM6 and Jurkat cells was not changed by incubation with neither TiAlV nor CoNiCrMo particles.
- “Interestingly, this study shows that the inflammasome can be exclu- 278 sively activated by CoNiCrMo but not TiAlV particles in osteoblasts. This is in contrast to 279 another study showing that titanium (Ti) particles were able to activate NLRP3 in- 280 fammasome components in macrophages”
– What we can observe in p20 Caspase 1 form figure 5 is, Ti group had upregulated protein at 10^7, than the CoNiCrMo groups, please reconsider this statement, as the results are suggesting the other way.
The reviewer is right that there is a dose dependent tendency towards an increase of p20 caspase-1 in MM6 with TiAlV particles. However, as there is no statistically significant effect, we would like to keep our conclusion as it is. The only significant effect on p20 caspase-1 has been observed for MG63 cells with CoNiCrMo particles.
Round 2
Reviewer 2 Report
The manuscript can be published in present form